# Scalable Inference for Gaussian Process Models with Black-Box Likelihoods

**Amir Dezfouli**
The University of New South Wales
akdezfuli@gmail.com

**Edwin V. Bonilla**
The University of New South Wales
e.bonilla@unsw.edu.au

## Abstract

We propose a sparse method for scalable automated variational inference (AVI) in a large class of models with Gaussian process (GP) priors, multiple latent functions, multiple outputs and non-linear likelihoods. Our approach maintains the statistical efficiency property of the original AVI method, requiring only expectations over univariate Gaussian distributions to approximate the posterior with a mixture of Gaussians. Experiments on small datasets for various problems including regression, classification, Log Gaussian Cox processes, and warped GPs show that our method can perform as well as the full method under high sparsity levels. On larger experiments using the MNIST and the SARCOS datasets we show that our method can provide superior performance to previously published scalable approaches that have been handcrafted to specific likelihood models.

## 1 Introduction

Developing automated yet practical approaches to Bayesian inference is a problem that has attracted considerable attention within the probabilisitic machine learning community (see e.g. [1, 2, 3, 4]). In the case of models with Gaussian process (GP) priors, the main challenge is that of dealing with a large number of highly-coupled latent variables. Although promising directions within the sampling community such as Elliptical Slice Sampling (ESS, [5]) have been proposed, they have been shown to be particularly slow compared to variational methods. In particular, [6] showed that their *automated variational inference* (AVI) method can provide posterior distributions that are practically indistinguishable from those obtained by ESS, while running orders of magnitude faster.

One of the fundamental properties of the method proposed in [6] is its *statistical efficiency*, which means that, in order to approximate a posterior distribution via the maximization of the evidence lower bound (ELBO), it only requires expectations over univariate Gaussian distributions regardless of the likelihood model. Remarkably, this property holds for a large class of models involving multiple latent functions and multiple outputs. However, this method is still impractical for large datasets as it inherits the cubic computational cost of GP models on the number of observations ($N$). While there have been several approaches to large scale inference in GP models [7, 8, 9, 10, 11], these have been focused on regression and classification problems. The main obstacle to apply these approaches to inference with general likelihood models is that it is unclear how they can be extended to frameworks such as those in [6], while maintaining that desirable property of statistical efficiency.

In this paper we build upon the inducing-point approach underpinning most sparse approximations to GPs [12, 13] in order to scale up the automated inference method of [6]. In particular, for models with multiple latent functions, multiple outputs and non-linear likelihoods (such as in multi-class classification and Gaussian process regression networks [14]) we propose a sparse approximation whose computational complexity is $\mathcal{O}(M^3)$ in time, where $M \ll N$ is the number of inducing points. This approximation maintains the statistical efficiency property of the original AVI method. As the resulting ELBO decomposes over the training data points, our method can scale up to a very

large number of observations and is amenable to stochastic optimization and parallel computation. Moreover, it can, in principle, approximate arbitrary posterior distributions as it uses a Mixture-of-Gaussians (MoG) as the family of approximate posteriors. We refer to our method as SAVIGP, which stands for *scalable automated variational inference for Gaussian process* models.

Our experiments on small datasets for problems including regression, classification, Log Gaussian Cox processes, and warped GPs [15] show that SAVIGP can perform as well as the full method under high levels of sparsity. On a larger experiment on the MNIST dataset, our approach outperforms the distributed variational inference method in [9], who used a class-conditional density modeling approach. Our method, unlike [9], uses a single discriminative multi-class framework. Finally, we use SAVIGP to do inference for the Gaussian process regression network model [14] on the SAR-COS dataset concerning an inverse robot dynamics problem [16]. We show that we can outperform previously published scalable approaches that used likelihood-specific inference algorithms.

## 2 Related work

There has been a long-standing interest in the GP community to overcome the cubic scaling of inference in standard GP models [17, 18, 12, 13, 8]. However, none of these approaches actually dealt with the harder tasks of developing scalable inference methods for multi-output problems and general likelihood models. The former (multiple output problem) has been addressed, notably, by [19] and [20] using the convolution process formalism. Nevertheless, such approaches were specific to regression problems. The latter problem (general likelihood models) has been tackled from a sampling perspective [5] and within an optimization framework using variational inference [21]. In particular, the work of [21] proposes an efficient full Gaussian posterior approximation for GP models with iid observations. Our work pushes this breakthrough further by allowing multiple latent functions, multiple outputs, and more importantly, scalability to large datasets.

A related area of research is that of modeling complex data with deep belief networks based on Gaussian process mappings [22]. Unlike our approach, these models target the unsupervised problem of discovering structure in high-dimensional data, do not deal with black-box likelihoods, and focus on small-data applications. Finally, very recent developments in speeding-up probabilistic kernel machines [9, 23, 24] show that the types of problems we are addressing here are highly relevant to the machine learning community. In particular, [23] has proposed efficient inference methods for large scale GP classification and [9] has developed a distributed variational approach for GP models, with a focus on regression and classification problems. Our work, unlike these approaches, allows practitioners and researchers to investigate new models with GP priors and complex likelihoods for which currently there is no machinery that can scale to very large datasets.

## 3 Gaussian Process priors and multiple-output nonlinear likelihoods

We are given a dataset $\mathcal{D} = \{\mathbf{x}_n, \mathbf{y}_n\}_{n=1}^N$, where $\mathbf{x}_n$ is a $D$-dimensional input vector and $\mathbf{y}_n$ is a $P$-dimensional output. Our goal is to learn the mapping from inputs to outputs, which can be established via $Q$ underlying latent functions $\{f_j\}_{j=1}^Q$. A sensible modeling approach to the above problem is to assume that the $Q$ latent functions $\{f_j\}$ are uncorrelated a priori and that they are drawn from $Q$ zero-mean Gaussian processes [25]:

$$p(\mathbf{f}) = \prod_{j=1}^Q p(\mathbf{f}_{\cdot j}) = \prod_{j=1}^Q \mathcal{N}(\mathbf{f}_{\cdot j}; \mathbf{0}, \mathbf{K}_j), \tag{1}$$

where $\mathbf{f}$ is the set of all latent function values; $\mathbf{f}_{\cdot j} = \{f_j(\mathbf{x}_n)\}_{n=1}^N$ denotes the values of latent function $j$; and $\mathbf{K}_j$ is the covariance matrix induced by the covariance function $\kappa_j(\cdot, \cdot)$, evaluated at every pair of inputs. Along with the prior in Equation (1), we can also assume that our multidimensional observations $\{\mathbf{y}_n\}$ are iid given the corresponding set of latent functions $\{\mathbf{f}_n\}$:

$$p(\mathbf{y}|\mathbf{f}) = \prod_{n=1}^N p(\mathbf{y}_n|\mathbf{f}_{n\cdot}), \tag{2}$$

where $\mathbf{y}$ is the set of all output observations; $\mathbf{y}_n$ is the $n$th output observation; and $\mathbf{f}_{n\cdot} = \{f_j(\mathbf{x}_n)\}_{j=1}^Q$ is the set of latent function values which $\mathbf{y}_n$ depends upon. In short, we are inter-

ested in models for which the following criteria are satisfied: (i) *factorization of the prior over the latent functions*; and (ii) *factorization of the conditional likelihood over the observations given the latent functions*. Interestingly, a large class of problems can be well modeled with the above assumptions: binary classification [7, 26], warped GPs [15], log Gaussian Cox processes [27], multi-class classification [26], and multi-output regression [14] all belong to this family of models.

## 3.1 Automated variational inference

One of the key inference challenges in the above models is that of computing the posterior distribution over the latent functions $p(\mathbf{f}|\mathbf{y})$. Ideally, we would like an efficient method that does not need to know the details of the likelihood in order to carry out posterior inference. This is exactly the main result in [6], which approximates the posterior with a mixture-of-Gaussians within a variational inference framework. This entails the optimization of an evidence lower bound, which decomposes as a KL-divergence term and an expected log likelihood (ELL) term. As the KL-divergence term is relatively straightforward to deal with, we focus on their main result regarding the ELL term:

**[6], Th. 1**: "*The expected log likelihood and its gradients can be approximated using samples from univariate Gaussian distributions*". More generally, we say that the ELL term and its gradients can be estimated using expectations over univariate Gaussian distributions. We refer to this result as that of *statistical efficiency*. One of the main limitations of this method is its poor scalability to large datasets, as it has a cubic time complexity on the number of data points, i.e. $\mathcal{O}(N^3)$. In the next section we describe our inference method that scales up to large datasets while maintaining the statistical efficiency property of the original model.

# 4 Scalable inference

In order to make inference scalable we redefine our prior to be sparse by conditioning the latent processes on a set of inducing variables $\{\mathbf{u}_{\cdot j}\}_{j=1}^{Q}$, which lie in the same space as $\{\mathbf{f}_{\cdot j}\}$ and are drawn from the same zero-mean GP priors. As before, we assume factorization of the prior across the $Q$ latent functions. Hence the resulting **sparse prior** is given by:

$$p(\mathbf{u}) = \prod_{j=1}^{Q} \mathcal{N}(\mathbf{u}_{\cdot j}; \mathbf{0}, \kappa(\mathbf{Z}_j, \mathbf{Z}_j)), \qquad p(\mathbf{f}|\mathbf{u}) = \prod_{j=1}^{Q} \mathcal{N}(\mathbf{f}_{\cdot j}; \tilde{\boldsymbol{\mu}}_j, \widetilde{\mathbf{K}}_j), \tag{3}$$

$$\tilde{\boldsymbol{\mu}}_j = \kappa(\mathbf{X}, \mathbf{Z}_j)\kappa(\mathbf{Z}_j, \mathbf{Z}_j)^{-1}\mathbf{u}_{\cdot j}, \tag{4}$$

$$\widetilde{\mathbf{K}}_j = \kappa_j(\mathbf{X}, \mathbf{X}) - \mathbf{A}_j\kappa(\mathbf{Z}_j, \mathbf{X}) \text{ with } \mathbf{A}_j = \kappa(\mathbf{X}, \mathbf{Z}_j)\kappa(\mathbf{Z}_j, \mathbf{Z}_j)^{-1}, \tag{5}$$

where $\mathbf{u}_{\cdot j}$ are the inducing variables for latent process $j$; $\mathbf{u}$ is the set of all the inducing variables; $\mathbf{Z}_j$ are all the inducing inputs (i.e. locations) for latent process $j$; $\mathbf{X}$ is the matrix of all input locations $\{\mathbf{x}_i\}$; and $\kappa(\mathbf{U}, \mathbf{V})$ is the covariance matrix induced by evaluating the covariance function $\kappa_j(\cdot, \cdot)$ at all pairwise vectors of matrices $\mathbf{U}$ and $\mathbf{V}$. We note that while each of the inducing variables in $\mathbf{u}_{\cdot j}$ lies in the same space as the elements in $\mathbf{f}_{\cdot j}$, each of the $M$ inducing inputs in $\mathbf{Z}_j$ lies in the same space as each input data point $\mathbf{x}_n$. Given the latent function values $\mathbf{f}_{n\cdot}$, the **conditional likelihood** factorizes across data points and is given by Equation (2).

## 4.1 Approximate posterior

We will approximate the posterior using variational inference. Motivated by the fact that the true joint posterior is given by $p(\mathbf{f}, \mathbf{u}|\mathbf{y}) = p(\mathbf{f}|\mathbf{u}, \mathbf{y})p(\mathbf{u}|\mathbf{y})$, our approximate posterior has the form:

$$q(\mathbf{f}, \mathbf{u}|\mathbf{y}) = p(\mathbf{f}|\mathbf{u})q(\mathbf{u}), \tag{6}$$

where $p(\mathbf{f}|\mathbf{u})$ is the conditional prior given in Equation (3) and $q(\mathbf{u})$ is our approximate (variational) posterior. This decomposition has proved effective in problems with a single latent process and a single output (see e.g. [13]).

Our variational distribution is a mixture of Gaussians (MoG):

$$q(\mathbf{u}|\boldsymbol{\lambda}) = \sum_{k=1}^{K} \pi_k q_k(\mathbf{u}|\mathbf{m}_k, \mathbf{S}_k) = \sum_{k=1}^{K} \pi_k \prod_{j=1}^{Q} \mathcal{N}(\mathbf{u}_{\cdot j}; \mathbf{m}_{kj}, \mathbf{S}_{kj}), \tag{7}$$

where $\boldsymbol{\lambda} = \{\pi_k, \mathbf{m}_{kj}, \mathbf{S}_{kj}\}$ are the variational parameters: the mixture proportions $\{\pi_k\}$, the posterior means $\{\mathbf{m}_{kj}\}$ and posterior covariances $\{\mathbf{S}_{kj}\}$ of the inducing variables corresponding to mixture component $k$ and latent function $j$. We also note that each of the mixture components $q_k(\mathbf{u}|\mathbf{m}_k, \mathbf{S}_k)$ is a Gaussian with mean $\mathbf{m}_k$ and block-diagonal covariance $\mathbf{S}_k$.

## 5 Posterior approximation via optimization of the evidence lower bound

Following variational inference principles, the log marginal likelihood $\log p(\mathbf{y})$ (or evidence) is lower bounded by the variational objective:

$$\log p(\mathbf{y}) \geq \mathcal{L}_{\mathrm{elbo}} = \underbrace{\int q(\mathbf{u}|\boldsymbol{\lambda})p(\mathbf{f}|\mathbf{u}) \log p(\mathbf{y}|\mathbf{f})\mathrm{d}\mathbf{f}\mathrm{d}\mathbf{u}}_{\mathcal{L}_{\mathrm{ell}}} \underbrace{-\mathrm{KL}(q(\mathbf{u}|\boldsymbol{\lambda})\|p(\mathbf{u}))}_{\mathcal{L}_{\mathrm{kl}}}, \tag{8}$$

where the evidence lower bound ($\mathcal{L}_{\mathrm{elbo}}$) decomposes as the sum of an expected log likelihood term ($\mathcal{L}_{\mathrm{ell}}$) and a KL-divergence term ($\mathcal{L}_{\mathrm{kl}}$). Our goal is to estimate our posterior distribution $q(\mathbf{u}|\boldsymbol{\lambda})$ via maximization of $\mathcal{L}_{\mathrm{elbo}}$. We consider first the $\mathcal{L}_{\mathrm{ell}}$ term, as it is the most difficult to deal with since we do not know the details of the implementation of the conditional likelihood $p(\mathbf{y}|\mathbf{f})$.

### 5.1 Expected log likelihood term

Here we need to compute the expectation of the log conditional likelihood $\log p(\mathbf{y}|\mathbf{f})$ over the joint approximate posterior given in Equation (6). Our goal is to obtain expressions for the $\mathcal{L}_{\mathrm{ell}}$ term and its gradients wrt the variational parameters while maintaining the statistical efficiency property of needing only expectations from univariate Gaussians. For this we first introduce an intermediate distribution $q(\mathbf{f}|\boldsymbol{\lambda})$ that is obtained by integrating out $\mathbf{u}$ from the joint approximate posterior:

$$\mathcal{L}_{\mathrm{ell}}(\boldsymbol{\lambda}) = \int_{\mathbf{f}} \int_{\mathbf{u}} q(\mathbf{u}|\boldsymbol{\lambda})p(\mathbf{f}|\mathbf{u}) \log p(\mathbf{y}|\mathbf{f})\mathrm{d}\mathbf{f}\mathrm{d}\mathbf{u} = \int_{\mathbf{f}} \log p(\mathbf{y}|\mathbf{f}) \underbrace{\int_{\mathbf{u}} p(\mathbf{f}|\mathbf{u})q(\mathbf{u}|\boldsymbol{\lambda})\mathrm{d}\mathbf{u}}_{q(\mathbf{f}|\boldsymbol{\lambda})} \mathrm{d}\mathbf{f}. \tag{9}$$

Given our approximate posterior in Equation (7), $q(\mathbf{f}|\boldsymbol{\lambda})$ can be obtained analytically:

$$q(\mathbf{f}|\boldsymbol{\lambda}) = \sum_{k=1}^{K} \pi_k q_k(\mathbf{f}|\boldsymbol{\lambda}_k) = \sum_{k=1}^{K} \pi_k \prod_{j=1}^{Q} \mathcal{N}(\mathbf{f}_{\cdot j}; \mathbf{b}_{kj}, \boldsymbol{\Sigma}_{kj}), \text{ with} \tag{10}$$

$$\mathbf{b}_{kj} = \mathbf{A}_j \mathbf{m}_{kj}, \qquad \boldsymbol{\Sigma}_{kj} = \widetilde{\mathbf{K}}_j + \mathbf{A}_j \mathbf{S}_{kj} \mathbf{A}_j^T, \tag{11}$$

where $\widetilde{\mathbf{K}}_j$ and $\mathbf{A}_j$ are given in Equation (5). Now we can rewrite Equation (9) as:

$$\mathcal{L}_{\mathrm{ell}}(\boldsymbol{\lambda}) = \sum_{k=1}^{K} \pi_k \mathbb{E}_{q_k(\mathbf{f}|\boldsymbol{\lambda}_k)}[\log p(\mathbf{y}|\mathbf{f})] = \sum_{n=1}^{N} \sum_{k=1}^{K} \pi_k \mathbb{E}_{q_{k(n)}(\mathbf{f}_{n\cdot})}[\log p(\mathbf{y}_{n\cdot}|\mathbf{f}_{n\cdot})], \tag{12}$$

where $\mathbb{E}_{q(x)}[g(x)]$ denotes the expectation of function $g(x)$ over the distribution $q(x)$. Here we have used the mixture decomposition of $q(\mathbf{f}|\boldsymbol{\lambda})$ in Equation (10) and the factorization of the likelihood over the data points in Equation (2). Now we are ready to state formally our main result.

**Theorem 1** *For the sparse GP model with prior defined in Equations (3) to (5), and likelihood defined in Equation (2), the expected log likelihood over the variational distribution in Equation (7) and its gradients can be estimated using expectations over univariate Gaussian distributions.*

Given the result in Equation (12), the proof is trivial for the computation of $\mathcal{L}_{\mathrm{ell}}$ as we only need to realize that $q_k(\mathbf{f}|\boldsymbol{\lambda}_k) = \mathcal{N}(\mathbf{f}; \mathbf{b}_k, \boldsymbol{\Sigma}_k)$ given in Equation (10) has a block-diagonal covariance structure. Consequently, $q_{k(n)}(\mathbf{f}_{n\cdot})$ is a $Q$-dimensional Gaussian with diagonal covariance. For the gradients of $\mathcal{L}_{\mathrm{ell}}$ wrt the variational parameters, we use the following identity:

$$\nabla_{\boldsymbol{\lambda}_k} \mathbb{E}_{q_{k(n)}(\mathbf{f}_{n\cdot})}[\log p(\mathbf{y}_n|\mathbf{f}_{n\cdot})] = \mathbb{E}_{q_{k(n)}(\mathbf{f}_{n\cdot})} \nabla_{\boldsymbol{\lambda}_k} \log q_{k(n)}(\mathbf{f}_{n\cdot}) \log p(\mathbf{y}_n|\mathbf{f}_{n\cdot}), \tag{13}$$

for $\boldsymbol{\lambda}_k \in \{\mathbf{m}_k, \mathbf{S}_k\}$, and the result for $\{\pi_k\}$ is straightforward. ∎

**Explicit computation of $\mathcal{L}_{\text{ell}}$**

We now provide explicit expressions for the computation of $\mathcal{L}_{\text{ell}}$. We know that $q_{k(n)}(\mathbf{f}_{n\cdot})$ is a $Q$-dimensional Gaussian with :

$$q_{k(n)}(\mathbf{f}_{n\cdot}) = \mathcal{N}(\mathbf{f}_{n\cdot}; \mathbf{b}_{k(n)}, \mathbf{\Sigma}_{k(n)}), \tag{14}$$

where $\mathbf{\Sigma}_{k(n)}$ is a diagonal matrix. The $j$th element of the mean and the $(j,j)$th entry of the covariance are given by:

$$[\mathbf{b}_{k(n)}]_j = [\mathbf{A}_j]_{n,:}\mathbf{m}_{kj}, \qquad [\mathbf{\Sigma}_{k(n)}]_{j,j} = [\widetilde{\mathbf{K}}_j]_{n,n} + [\mathbf{A}_j]_{n,:}\mathbf{S}_{kj}[\mathbf{A}_j^T]_{:,n}, \tag{15}$$

where $[\mathbf{A}]_{n,:}$ and $[\mathbf{A}]_{:,n}$ denote the $n$th row and $n$th column of matrix $\mathbf{A}$ respectively. Hence we can compute $\mathcal{L}_{\text{ell}}$ as follows:

$$\left\{\mathbf{f}_{n\cdot}^{(k,i)}\right\}_{i=1}^S \sim \mathcal{N}(\mathbf{f}_{n\cdot}; \mathbf{b}_{k(n)}, \mathbf{\Sigma}_{k(n)}), k = 1, \ldots, K, \tag{16}$$

$$\widehat{\mathcal{L}}_{\text{ell}} = \frac{1}{S}\sum_{n=1}^N \sum_{k=1}^K \pi_k \sum_{i=1}^S \log p(\mathbf{y}_{n\cdot}|\mathbf{f}_{n\cdot}^{(k,i)}). \tag{17}$$

The gradients of $\mathcal{L}_{\text{ell}}$ wrt variational parameters are given in the supplementary material.

## 5.2 KL-divergence term

We turn now our attention to the KL-divergence term, which can be decomposed as follows:

$$-\text{KL}(q(\mathbf{u}|\boldsymbol{\lambda})\|p(\mathbf{u})) = \underbrace{\mathbb{E}_q[-\log q(\mathbf{u}|\boldsymbol{\lambda})]}_{\mathcal{L}_{\text{ent}}} + \underbrace{\mathbb{E}_q[\log p(\mathbf{u})]}_{\mathcal{L}_{\text{cross}}}, \tag{18}$$

where the entropy term ($\mathcal{L}_{\text{ent}}$) can be lower bounded using Jensen's inequality:

$$\mathcal{L}_{\text{ent}} \geq -\sum_{k=1}^K \pi_k \log \sum_{\ell=1}^K \pi_\ell \mathcal{N}(\mathbf{m}_k; \mathbf{m}_\ell, \mathbf{S}_k + \mathbf{S}_\ell) \stackrel{\text{def}}{=} \hat{\mathcal{L}}_{\text{ent}}. \tag{19}$$

The negative cross-entropy term ($\mathcal{L}_{\text{cross}}$) can be computed exactly:

$$\mathcal{L}_{\text{cross}} = -\frac{1}{2}\sum_{k=1}^K \pi_k \sum_{j=1}^Q [M \log 2\pi + \log|\kappa(\mathbf{Z}_j, \mathbf{Z}_j)| + \mathbf{m}_{kj}^T \kappa(\mathbf{Z}_j, \mathbf{Z}_j)^{-1}\mathbf{m}_{kj} + \text{tr}\,\kappa(\mathbf{Z}_j, \mathbf{Z}_j)^{-1}\mathbf{S}_{kj}]. \tag{20}$$

The gradients of the above terms wrt the variational parameters are given in the supplementary material.

## 5.3 Hyperparameter learning and scalability to large datasets

For simplicity in the notation we have omitted the parameters of the covariance functions and the likelihood parameters from the ELBO. However, in our experiments we optimize these along with the variational parameters in a variational-EM alternating optimization framework. The gradients of the ELBO wrt these parameters are given in the supplementary material.

The original framework of [6] is completely unfeasible for large datasets, as its complexity is dominated by the inversion of the Gram matrix on all the training data, which is an $\mathcal{O}(N^3)$ operation where $N$ is the number of training points. Our sparse framework makes automated variational inference practical for large datasets as its complexity is dominated by inversions of the kernel matrix on the inducing points, which is an $\mathcal{O}(M^3)$ operation where $M$ is the number of inducing points per latent process. Furthermore, as the $\mathcal{L}_{\text{ell}}$ and its gradients decompose over the training points, and the $\mathcal{L}_{\text{kl}}$ term decomposes over the number of latent process, our method is amenable to stochastic optimization and / or parallel computation, which makes it scalable to very large number of input observations, output dimensions and latent processes. In our experiments in section 6 we show that our sparse framework can achieve similar performance to the full method [6] on small datasets under high levels of sparsity. Moreover, we carried out experiments on larger datasets for which is practically impossible to apply the full (i.e. non-sparse) method.

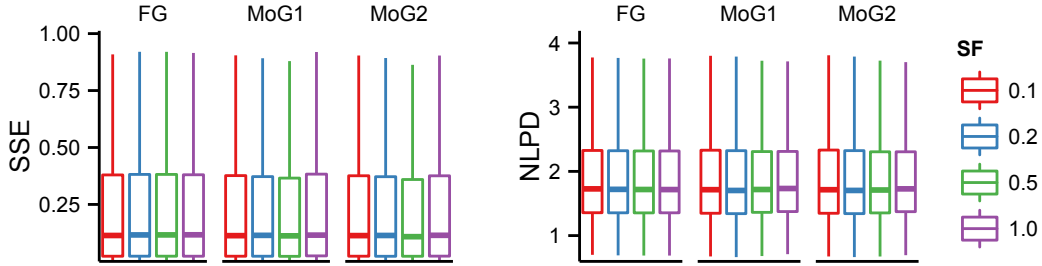

Figure 1: The SSE and NLPD for *warped GPs* on the Abalone dataset, where lower values on both measures are better. Three approximate posteriors are used: FG (full Gaussian), MoG1 (diagonal Gaussian), and MoG2 (mixture of two diagonal Gaussians), along with various sparsity factors (SF = M/N). The smaller the SF the sparser the model, with SF=1 corresponding to no sparsity.

## 6 Experiments

Our experiments first consider the same six benchmarks with various likelihood models analyzed by [6]. The number of training points ($N$) on these benchmarks ranges from 300 to 1233 and their input dimensionality ($D$) ranges from 1 to 256. The goal of this first set of experiments is to show that SAVIGP can attain as good performance as the full method under high sparsity levels. We also carried out experiments at a larger scale using the MNIST dataset and the SARCOS dataset [16]. The application of the original automated variational inference framework on these datasets is unfeasible. We refer the reader to the supplementary material for the details of our experimental set-up.

We used two performance measures in each experiment: the standardized squared error (SSE) and the negative log predictive density (NLPD) for continuous-output problems, and the error rate and the negative log probability (NLP) for discrete-output problems. We use three versions of SAVIGP: FG, MoG1, and MoG2, corresponding to a full Gaussian, a diagonal Gaussian, and mixture of diagonal Gaussians with 2 components, respectively. We refer to the ratio of the number of inducing points over the number of training points ($M/N$) as *sparsity factor*.

### 6.1 Small-scale experiments

In this section we describe the results on three (out of six) benchmarks used by [6] and analyze the performance of SAVIGP. The other three benchmarks are described in the supplementary material.

**Warped Gaussian processes (WGP)**, Abalone dataset [28], $p(y_n|f_n) = \nabla_{y_n} t(y_n) \mathcal{N}(t(y_n)|f_n, \sigma^2)$. For this task we used the same neural-net transformation as in [15] and the results for the Abalone dataset are shown in Figure 1. We see that the performance of SAVIGP is practically indistinguishable across all sparsity factors for SSE and NLPD. Here we note that [6] showed that automated variational inference performed competitively when compared to hand-crafted methods for warped GPs [15].

**Log Gaussian Cox process (LGCP)**, Coal-mining disasters dataset [29], $p(y_n|f_n) = \frac{\lambda_n^{y_n} \exp(-\lambda_n)}{y_n!}$. Here we used the LGCP for modeling the number of coal-mining disasters between years 1851 to 1962. We note that [6] reported that automated variational inference (the focus of this paper) produced practically indistinguishable distributions (but run order of magnitude faster) when compared to sampling methods such as Elliptical Slice Sampling [5]. The results for our sparse models are shown in Figure 2, where we see that both models (FG and MoG1) remain mostly unaffected when using high levels of sparsity. We also confirm the findings in [6] that the MoG1 model underestimates the variance of the predictions.

**Binary classification**, Wisconsin breast cancer dataset [28], $p(y_n = 1) = 1/(1 + \exp(-f_n))$. Classification error rates and the negative log probability (NLP) on the Wisconsin breast cancer dataset are shown in Figure 3. We see that the error rates are comparable across all models and sparsity factors. Interestingly, sparser models achieved lower NLP values, suggesting overconfident predictions by the less sparse models, especially for the mixtures of diagonal Gaussians.

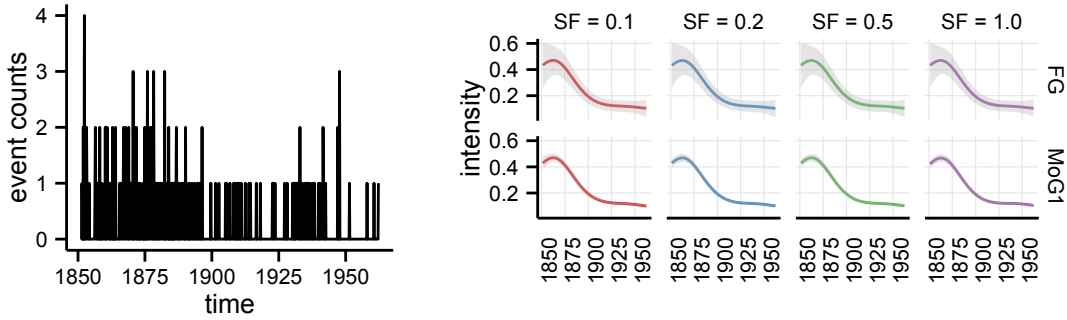

Figure 2: Left: the coal-mining disasters data. Right: the posteriors for a *Log Gaussian Cox process* on these data when using a full Gaussian (FG) and a diagonal Gaussian (MoG1), for various sparsity factors (SF = M/N). The smaller the SF the sparser the model, with SF=1 corresponding to no sparsity. The solid line is the posterior mean and the shading area includes 90% confidence interval.

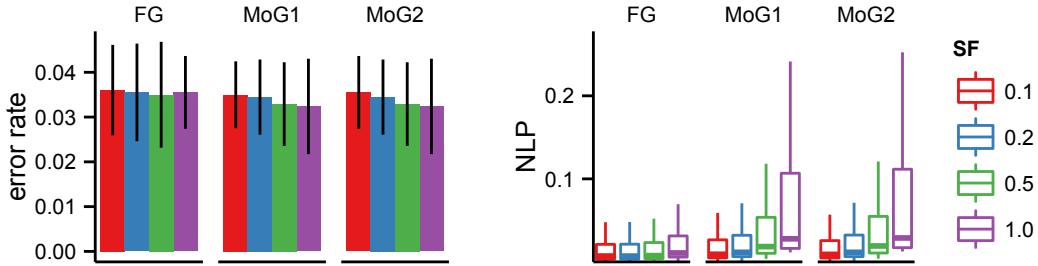

Figure 3: Error rates and NLP for *binary classification* on the Wisconsin breast cancer dataset. Three approximate posteriors are used: FG (full Gaussian), MoG1 (diagonal Gaussian), and MoG2 (mixture of two diagonal Gaussians), along with various sparsity factors (SF = M/N). The smaller the SF the sparser the model, with SF=1 corresponding to the original model without sparsity. Error bars on the left plot indicate 95% confidence interval around the mean.

## 6.2 Large-scale experiments

In this section we show the results of the experiments carried out on larger datasets with non-linear non-Gaussian likelihoods.

**Multi-class classification on the MNIST dataset**. We first considered a multi-class classification task on the MNIST dataset using the softmax likelihood. This dataset has been extensively used by the machine learning community and contains 50,000 examples for training, 10,000 for validation and 10,000 for testing, with 784-dimensional input vectors. Unlike most previous approaches, we did not tune additional parameters using the validation set. Instead we used our variational framework for learning all the model parameters using all the training and validation data. This setting most likely provides a lower bound on test accuracy but our goal here is simply to show that we can achieve competitive performance with highly-sparse models as our inference algorithm does not know the details of the conditional likelihood. Figure 4 (left and middle) shows error rates and NLPs where we see that, although the performance decreases with sparsity, the method is able to attain an accuracy of 97.49%, while using only around 2000 inducing points (SF = 0.04).

To the best of our knowledge, we are the first to train a multi-class Gaussian process classifier using a single discriminative probabilistic framework on all classes on MNIST. For example, [17] used a 1-vs-rest approach and [23] focused on the binary classification task of distinguishing the odd digits from the even digits. Finally, [9] trained one model for each digit and used it as a density model, achieving an error rate of 5.95%. Our experiments show that by having a single discriminative probabilistic framework, even without exploiting the details of the conditional likelihood, we can bring this error rate down to 2.51%. As a reference, previous literature reports about 12% error rate by linear classifiers and less than 1% error rate by sate-of-the-art large/deep convolutional nets.

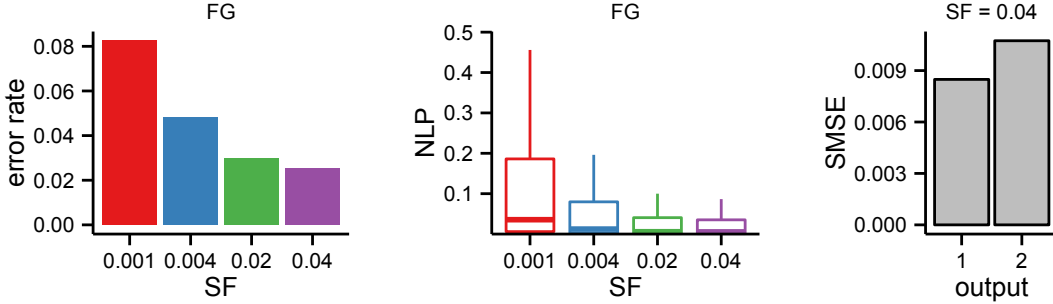

Figure 4: Left and middle: classification error rates and negative log probabilities (NLP) for the *multi-class problem* on MNIST. Here we used the FG (full Gaussian) approximation with various sparsity factors (SF = M/N). The smaller the SF the sparser the model. Right: the SMSE for a *Gaussian process regression network* model on the SARCOS dataset when learning the 4th and 7th torques (output 1 and output 2) with a FG (full Gaussian) approximation and 0.04 sparsity factor.

Our results show that our method, while solving the harder problem of full posterior estimation, can reduce the gap between GPs and deep nets.

**Gaussian process regression networks on the SARCOS dataset**. Here we apply our SAVIGP inference method to the Gaussian process regression networks (GPRNs) model of [14], using the SARCOS dataset as a test bed. GPRNs are a very flexible regression approach where $P$ outputs are a linear combination of $Q$ latent Gaussian processes, with the weights of the linear combination also drawn from Gaussian processes. This yields a non-linear multiple output likelihood model where the correlations between the outputs can be spatially adaptive, i.e. input dependent. The SARCOS dataset concerns an inverse dynamics problem of a 7-degrees-of-freedom anthropomorphic robot arm [16]. The data consists of 44,484 training examples mapping from a 21-dimensional input space (7 joint positions, 7 joint velocities, 7 joint accelerations) to the corresponding 7 joint torques. Similarly to the work in [10], we consider joint learning for the 4th and 7th torques, which we refer to as output 1 and output 2 respectively, and make predictions on 4,449 test points per output.

Figure 4 (right) shows the standardized mean square error (SMSE) with the full Gaussian approximation (FG) using SF=0.04, i.e. less than 2000 inducing points. The results are considerably better than those reported by [10] (0.2631 and 0.0127 for each output respectively), although their setting was much sparser than ours on the first output. This also corroborates previous findings that, on this problem, having more data does help [16]. To the best of our knowledge, we are the first to perform inference in GPRNs on problems at this scale.

## 7 Conclusion

We have presented a scalable approximate inference method for models with Gaussian process (GP) priors, multiple outputs, and nonlinear likelihoods. One of the key properties of this method is its *statistical efficiency* in that it requires only expectations over univariate Gaussian distributions to approximate the posterior with a mixture of Gaussians. Extensive experimental evaluation shows that our approach can attain excellent performance under high sparsity levels and that it can outperform previous inference methods that have been handcrafted to specific likelihood models. Overall, this work makes a substantial contribution towards the goal of developing generic yet scalable Bayesian inference methods for models based on Gaussian processes.

**Acknowledgments**

This work has been partially supported by UNSW's Faculty of Engineering Research Grant Program project # PS37866 and an AWS in Education Research Grant award. AD was also supported by a grant from the Australian Research Council # DP150104878.

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
