[Supplementary Material · supplementary-savigp-nips.pdf]

# Scalable Inference for Gaussian Process Models with Black-Box Likelihoods
## *Supplementary Material*

**Amir Dezfouli**
The University of New South Wales
akdezfuli@gmail.com

**Edwin V. Bonilla**
The University of New South Wales
e.bonilla@unsw.edu.au

## 1 Gradients of the evidence lower bound wrt variational parameters

Here we specify the gradients of the ELBO wrt variational parameters. For the covariance, we consider $\mathbf{S}_{kj}$ of general structure but also give the updates when $\mathbf{S}_{kj}$ is a diagonal matrix, denoted with $\tilde{\mathbf{S}}_{kj}$.

Let $\mathbf{K}_{zz}$ be the block-diagonal covariance with $Q$ blocks $\kappa(\mathbf{Z}_j, \mathbf{Z}_j)$, $j = 1, \dots Q$. Additionally, lets assume the following definitions:

$$\mathbf{C}_{kl} \overset{\text{def}}{=} \mathbf{S}_k + \mathbf{S}_\ell, \tag{1}$$

$$\mathcal{N}_{k\ell} \overset{\text{def}}{=} \mathcal{N}(\mathbf{m}_k; \mathbf{m}_\ell, \mathbf{C}_{kl}), \tag{2}$$

$$z_k \overset{\text{def}}{=} \sum_{\ell=1}^{K} \pi_\ell \mathcal{N}_{k\ell}. \tag{3}$$

The gradients of $\mathcal{L}_{\text{kl}}$ wrt the posterior mean and posterior covariance for component $k$ are:

$$\nabla_{\mathbf{m}_k} \mathcal{L}_{\text{cross}} = -\pi_k \mathbf{K}_{zz}^{-1} \mathbf{m}_k, \tag{4}$$

$$\nabla_{\mathbf{S}_k} \mathcal{L}_{\text{cross}} = -\frac{1}{2} \pi_k \mathbf{K}_{zz}^{-1}, \text{ and for diagonal covariance we have:} \tag{5}$$

$$\nabla_{\tilde{\mathbf{S}}_k} \mathcal{L}_{\text{cross}} = -\frac{1}{2} \pi_k \, \text{diag}(\mathbf{K}_{zz}^{-1}), \tag{6}$$

$$\nabla_{\pi_k} \mathcal{L}_{\text{cross}} = -\frac{1}{2} \sum_{j=1}^{Q} [M \log 2\pi + \log |\kappa(\mathbf{Z}_j, \mathbf{Z}_j)| + \mathbf{m}_{kj}^T \kappa(\mathbf{Z}_j, \mathbf{Z}_j)^{-1} \mathbf{m}_{kj} + \text{tr} \, \kappa(\mathbf{Z}_j, \mathbf{Z}_j)^{-1} \mathbf{S}_{kj}], \tag{7}$$

where we note that we compute $\mathbf{K}_{zz}^{-1}$ by inverting the corresponding blocks $\kappa(\mathbf{Z}_j, \mathbf{Z}_j)$ independently. The gradients of the entropy term wrt the variational parameters are:

$$\nabla_{\mathbf{m}_k} \hat{\mathcal{L}}_{\text{ent}} = \pi_k \sum_{\ell=1}^{K} \pi_\ell \left( \frac{\mathcal{N}_{k\ell}}{z_k} + \frac{\mathcal{N}_{k\ell}}{z_\ell} \right) \mathbf{C}_{kl}^{-1}(\mathbf{m}_k - \mathbf{m}_\ell), \tag{8}$$

$$\nabla_{\mathbf{S}_k} \hat{\mathcal{L}}_{\text{ent}} = \frac{1}{2} \pi_k \sum_{\ell=1}^{K} \pi_\ell \left( \frac{\mathcal{N}_{k\ell}}{z_k} + \frac{\mathcal{N}_{k\ell}}{z_\ell} \right) \left[ \mathbf{C}_{kl}^{-1} - \mathbf{C}_{kl}^{-1}(\mathbf{m}_k - \mathbf{m}_\ell)(\mathbf{m}_k - \mathbf{m}_\ell)^T \mathbf{C}_{kl}^{-1} \right], \tag{9}$$

and for diagonal covariance we have:

$$\nabla_{\tilde{\mathbf{S}}_k} \hat{\mathcal{L}}_{\text{ent}} = \frac{1}{2} \pi_k \sum_{\ell=1}^{K} \pi_\ell \left( \frac{\mathcal{N}_{k\ell}}{z_k} + \frac{\mathcal{N}_{k\ell}}{z_\ell} \right) \left[ \tilde{\mathbf{C}}_{kl}^{-1} - \tilde{\mathbf{C}}_{kl}^{-1} \, \text{diag}\left((\mathbf{m}_k - \mathbf{m}_\ell) \odot (\mathbf{m}_k - \mathbf{m}_\ell)\right) \tilde{\mathbf{C}}_{kl}^{-1} \right], \tag{10}$$

$$\nabla_{\pi_k} \hat{\mathcal{L}}_{\text{ent}} = -\log z_k - \sum_{\ell=1}^{K} \pi_\ell \frac{\mathcal{N}_{k\ell}}{z_\ell}, \tag{11}$$

where $\tilde{\mathbf{C}}_{kl}$ is the diagonal matrix defined analogously to $\mathbf{C}_{kl}$ in Equation (1). The gradients of the expected log likelihood term are:

$$\nabla \mathbf{m}_{kj}\widehat{\mathcal{L}}_{\text{ell}} = \frac{\pi_k}{S}\kappa(\mathbf{Z}_j, \mathbf{Z}_j)^{-1}\sum_{n=1}^{N}\kappa(\mathbf{Z}_j, \mathbf{x}_n)[\boldsymbol{\Sigma}_{k(n)}]_{j,j}^{-1}\sum_{i=1}^{S}\left(f_{nj}^{(k,i)} - [\mathbf{b}_{k(n)}]_j\right)\log p(\mathbf{y}_{n\cdot}|\mathbf{f}_{n\cdot}^{(k,i)}), \tag{12}$$

$$\nabla \mathbf{S}_{kj}\widehat{\mathcal{L}}_{\text{ell}} = \frac{\pi_k}{2S}\sum_{n=1}^{N}\left([\mathbf{A}_j^T]_{:,n}[\mathbf{A}_j]_{n,:}\right)\sum_{i=1}^{S}\left[[\boldsymbol{\Sigma}_{k(n)}]_{j,j}^{-2}\left(f_{nj}^{(k,i)} - [\mathbf{b}_{k(n)}]_j\right)^2 - [\boldsymbol{\Sigma}_{k(n)}]_{j,j}^{-1}\right]\log p(\mathbf{y}_{n\cdot}|\mathbf{f}_{n\cdot}^{(k,i)}), \tag{13}$$

$$\nabla \pi_k\widehat{\mathcal{L}}_{\text{ell}} = \frac{1}{S}\sum_{n=1}^{N}\sum_{i=1}^{S}\log p(\mathbf{y}_{n\cdot}|\mathbf{f}_{n\cdot}^{(k,i)}), \tag{14}$$

and for diagonal covariance $\tilde{\mathbf{S}}_{kj}$ we replace $\left([\mathbf{A}_j^T]_{:,n}[\mathbf{A}_j]_{n,:}\right)$ with $\text{diag}\left([\mathbf{A}_j^T]_{:,n}\odot[\mathbf{A}_j^T]_{:,n}\right)$ in Equation (13).

## 2  Gradients of the evidence lower bound wrt hyperparameters

The gradients wrt a covariance hyperparameter $\theta_j$ are:

$$\nabla_{\theta_j}\mathcal{L}_{\text{cross}} = -\frac{1}{2}\sum_{k=1}^{K}\pi_k\,\text{tr}\left[\kappa(\mathbf{Z}_j, \mathbf{Z}_j)^{-1}\nabla_{\theta_j}\kappa(\mathbf{Z}_j, \mathbf{Z}_j)\right. \tag{15}$$
$$\left. -\kappa(\mathbf{Z}_j, \mathbf{Z}_j)^{-1}\nabla_{\theta_j}\kappa(\mathbf{Z}_j, \mathbf{Z}_j)\kappa(\mathbf{Z}_j, \mathbf{Z}_j)^{-1}\left(\mathbf{m}_{kj}\mathbf{m}_{kj}^T + \mathbf{S}_j\right)\right].$$

For the $\widehat{\mathcal{L}}_{\text{ell}}$ we have that:

$$\nabla_{\theta_j}\widehat{\mathcal{L}}_{\text{ell}} = \sum_{n=1}^{N}\sum_{k=1}^{K}\pi_k\mathbb{E}_{q_{k(n)}(\mathbf{f}_{n\cdot})}\nabla_{\theta_j}\log q_{k(n)}(\mathbf{f}_{n\cdot})\log p(\mathbf{y}_{n\cdot}|\mathbf{f}_{n\cdot}), \tag{16}$$

and computing the corresponding gradient we obtain:

$$\nabla_{\theta_j}\widehat{\mathcal{L}}_{\text{ell}} = -\frac{1}{2}\sum_{n=1}^{N}\sum_{k=1}^{K}\pi_k\mathbb{E}_{q_{k(n)}(\mathbf{f}_{n\cdot})}\left([\boldsymbol{\Sigma}_{k(n)}]_{j,j}^{-1}\nabla_{\theta_j}[\boldsymbol{\Sigma}_{k(n)}]_{j,j} - 2(f_{nj} - [\mathbf{b}_{k(n)}]_j)[\boldsymbol{\Sigma}_{k(n)}]_{j,j}^{-1}\nabla_{\theta_j}[\mathbf{b}_{k(n)}]_j\right. \tag{17}$$
$$\left. - (f_{nj} - [\mathbf{b}_{k(n)}]_j)^2[\boldsymbol{\Sigma}_{k(n)}]_{j,j}^{-2}\nabla_{\theta_j}[\boldsymbol{\Sigma}_{k(n)}]_{j,j}\right)\log p(\mathbf{y}_{n\cdot}|\mathbf{f}_{n\cdot}),$$

for which we need:

$$\nabla_{\theta_j}[\mathbf{b}_{k(n)}]_j = \left(\nabla_{\theta_j}[\mathbf{A}_j]_{n,:}\right)\mathbf{m}_{kj}, \tag{18}$$

$$\nabla_{\theta_j}[\boldsymbol{\Sigma}_{k(n)}]_{j,j} = \nabla_{\theta_j}[\widetilde{\mathbf{K}}_j]_{n,n} + 2\left(\nabla_{\theta_j}[\mathbf{A}_j]_{n,:}\right)\mathbf{S}_{kj}[\mathbf{A}_j^T]_{:,n} \tag{19}$$

$$= \nabla_{\theta_j}\kappa(\mathbf{x}_n, \mathbf{x}_n) - \left(\nabla_{\theta_j}[\mathbf{A}_j]_{n,:}\right)\kappa(\mathbf{Z}_j, \mathbf{x}_n) - [\mathbf{A}_j]_{n,:}\nabla_{\theta_j}\kappa(\mathbf{Z}_j, \mathbf{x}_n) \tag{20}$$

$$+ 2\left(\nabla_{\theta_j}[\mathbf{A}_j]_{n,:}\right)\mathbf{S}_{kj}[\mathbf{A}_j^T]_{:,n}, \tag{21}$$

where

$$\nabla_{\theta_j}[\mathbf{A}_j]_{n,:} = \left(\nabla_{\theta_j}\kappa(\mathbf{x}_n, \mathbf{Z}_j) - [\mathbf{A}_j]_{n,:}\nabla_{\theta_j}\kappa(\mathbf{Z}_j, \mathbf{Z}_j)\right)\kappa(\mathbf{Z}_j, \mathbf{Z}_j)^{-1}. \tag{22}$$

## 3  Control Variates

We use control variates (see e.g. section 3.2 of [1]) to reduce the variance of the gradient estimates. In particular, we are interested in estimating gradients of the form:

$$\nabla_{\lambda_k}\mathbb{E}_{q_{k(n)}(\mathbf{f}_{n\cdot})}[\log p(\mathbf{y}_n|\mathbf{f}_{n\cdot})] = \mathbb{E}_{q_{k(n)}(\mathbf{f}_{n\cdot})}[g(\mathbf{f}_{n\cdot})], \text{ with} \tag{23}$$

$$g(\mathbf{f}_{n\cdot}) = \nabla_{\lambda_k}\log q_{k(n)}(\mathbf{f}_{n\cdot})\log p(\mathbf{y}_n|\mathbf{f}_{n\cdot}), \tag{24}$$

where the expectations are computed using samples from $q_{k(n)}(\mathbf{f}_{n\cdot})$, which depends on the variational parameter $\lambda_k$. As suggested by [1], a sensible control variate is the so-called score function

$$h(\mathbf{f}_{n\cdot}) = \nabla_{\lambda_k}\log q_{k(n)}(\mathbf{f}_{n\cdot}), \tag{25}$$

Figure 1: The distributions of SSE and NLPD for a regression task on the Boston housing dataset, where lower values on both performance measures are better. Three approximate posteriors in SAVIGP are used: FG (full Gaussian), MoG1 (diagonal Gaussian), and MoG2 (mixture of two diagonal Gaussians), along with various sparsity factors (SF = M/N). The smaller the SF the sparser the model, with SF=1 corresponding to the original model without sparsity.

whose expectation is zero. Hence, the function:

$$\tilde{g}(\mathbf{f}_{n\cdot}) = g(\mathbf{f}_{n\cdot}) - \hat{a}h(\mathbf{f}_{n\cdot}), \tag{26}$$

has the same expectation as $g(\mathbf{f}_{n\cdot})$ but lower variance when $\hat{a}$ is given by:

$$\hat{a} = \frac{\mathbb{C}\text{ov}[g(\mathbf{f}_{n\cdot}), h(\mathbf{f}_{n\cdot})]}{\mathbb{V}[h(\mathbf{f}_{n\cdot})]}, \tag{27}$$

where $\mathbb{C}\text{ov}[g(\mathbf{f}_{n\cdot}), h(\mathbf{f}_{n\cdot})]$ is the covariance between $g(\mathbf{f}_{n\cdot})$ and $h(\mathbf{f}_{n\cdot})$; $\mathbb{V}[h(\mathbf{f}_{n\cdot})]$ is the variance of $h(\mathbf{f}_{n\cdot})$; and both are estimated using samples from $q_{k(n)}(\mathbf{f}_{n\cdot})$. Therefore, our corrected gradient is given by:

$$\tilde{\nabla}_{\lambda_k}\mathbb{E}_{q_{k(n)}(\mathbf{f}_{n\cdot})}[\log p(\mathbf{y}_n|\mathbf{f}_{n\cdot})] \stackrel{\text{def}}{=} \mathbb{E}_{q_{k(n)}(\mathbf{f}_{n\cdot})}[\tilde{g}(\mathbf{f}_{n\cdot})] \tag{28}$$

$$= \mathbb{E}_{q_{k(n)}(\mathbf{f}_{n\cdot})}[\nabla_{\lambda_k}\log q_{k(n)}(\mathbf{f}_{n\cdot})(\log p(\mathbf{y}_n|\mathbf{f}_{n\cdot}) - \hat{a})]. \tag{29}$$

## 4  Details of experimental set-up

We used the squared exponential covariance function with automatic relevance determination (ARD) for Boston housing and the Creep dataset (see below). In all the other experiments the isotropic version was used. In the general case, a model contains three sets of parameters (variational parameters, covariance hyperparameters and likelihood parameters) that were estimated within an alternating optimization framework. Each set of parameters were optimized separately while keeping the rest of parameters fixed, and this was repeated until convergence for a maximum number of global iterations, which was set to 200. For the small experiments, each set of parameters were optimized for a maximum of 25 function evaluations. For the large-scale experiments, this maximum was set to 50 for the variational parameters, and 10 for the covariance hyperparameters and likelihood parameters. For the small experiments convergence was determined when changes in the ELBO were less than $1e$-5 or changes in the variational parameters were less than $1e$-3. For the large experiments, changes in the ELBO less than 10 was determined as convergence criterion. For all the methods we used the L-BFGS optimizer.

We used 2000 samples for approximating the ELBO and its gradients. Out of these samples, 200 samples were used to reduce the variance of the gradient estimators using the control variates described in section 3. For the small experiments, the inducing inputs were placed at a subset of the training data in a nested fashion so that experiments on less sparse models contained the inducing points of the sparser models. For the large experiments, inducing points were placed using k-means clustering.

## 5  Additional results on small datasets

**Standard regression**, Boston housing dataset [2], $p(y_n|f_n) = \mathcal{N}(y_n; f_n, \sigma^2)$. Figure 1 shows the results for the Boston housing dataset, where we see that all the models have similar SSE distributions under different levels of sparsity. Although for NLPD there is a trend towards better performance when using more inducing points, the performance is comparable across different sparsity factors. In particular, For MoG1 and MoG2 it seems that more sparsity makes the model to be less certain about the wrong predictions, as indicated by the smaller upper whiskers for lower values of SF in the NLPD panel.

**Warped Gaussian processes (WGP)**, Creep dataset [2], $p(y_n|f_n) = \nabla_{y_n}t(y_n)\mathcal{N}(t(y_n)|f_n, \sigma^2)$. For the Creep dataset (Figure 2) we note a small detriment in performance for the sparser models, although the SSE and the NLPD are still comparable across different levels of sparsity. Here we note that [3] showed that automated variational inference performed competitively when compared to hand-crafted methods for warped GPs [4].

**Multi-class classification**, USPS dataset [5], $p(y_n = c) = \exp(f_n^c)/\sum_{i=1}^{C}\exp(f_n^i)$. For this dataset we addressed the problem of a multi-class classification for the digits 4, 7, and 9, using the softmax likelihood. The classification error rates and

Figure 2: The distributions of SSE and NLPD for warped GPs on the Creep dataset, where lower values on both performance measures are better. Three approximate posteriors in SAVIGP are used: FG (full Gaussian), MoG1 (diagonal Gaussian), and MoG2 (mixture of two diagonal Gaussians), along with various sparsity factors (SF = M/N). The smaller the SF the sparser the model, with SF=1 corresponding to the original model without sparsity.

Figure 3: Classification error rates and negative log probabilities (NLP) for the multi-class classification problem on the USPS dataset. Three approximate posteriors in SAVIGP are used: FG (full Gaussian), MoG1 (diagonal Gaussian), and MoG2 (mixture of two diagonal Gaussians), along with various sparsity factors (SF = M/N). The smaller the SF the sparser the model, with SF=1 corresponding to the original model without sparsity. Error bars indicate 95% confidence interval around the mean.

the NLP in Figure 3 show a similar trend to the binary classification problem, with error rates comparable across models and sparsity factors, the NLP being improved by the sparser models, and the mixture of diagonal models with less sparsity being highly penalized for overconfident predictions.

# 6    Additional experiment on MNIST

In this section we present results on the task of distinguishing the odd digits from the even digits on the MNIST dataset. This task was also considered by [6]. On this task our method attains an accuracy of 97.8% and an NLP of 0.068, which are similar to those those reported by [6], 97.8% and 0.069, respectively. Figure 4 shows the error rate (ER) and the negative log probability (NLP) on the binary classification task of distinguishing the odd digits from the even digits on the MNIST dataset as a function of training time. We observe a rapid decrease of the error rate from around 50% to 10% in the first 1000 seconds and then converging to near best predictions soon after 2000 seconds. This shows that our method converges quickly to good solutions. The NLP decreases at a slower rate at the beginning but near-best values are also attained soon after 2000 seconds.

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

Figure 4:   The error rate (ER) and the negative log probability (NLP) on the binary classification task of distinguishing the odd digits from the even digits on the MNIST dataset as a function of training time.