[Reviews · NeurIPS 2015]

Submitted by Assigned_Reviewer_1

Are the authors also optimizing the locations of the inducing points?
Summary: Good paper in which automatic methods for VI with Gaussian processes are scaled to large datasets.

Submitted by Assigned_Reviewer_2

The paper marries a sparse approximation to the black-box inference method of [6], and demonstrates its effectiveness on a pleasingly wide variety of problems.

I almost can't believe it took us this long for this paper to be written, but I'm happy that it has.

The title was confusing.

It made sense once I realized it was referring to [6], but I wonder if there isn't a better word than "automated" to describe the fact that it can handle black-box likelihood functions.

Quality and clarity:

The intro was fairly clearly written, although I don't understand why the term "statistical efficiency" is used to describe a bound being well-approximated.

Shouldn't statistical efficiency refer to taking full advantage of the available data?

How can the computational complexity ( O(M^3)) not depend on the number of datapoints?

It's not clear that extending to "multiple latent functions" is much of a contribution, since it's not mathematically distinct from modeling a single function that's been broken into independent sections.

But perhaps it's worth pointing out anyways.

The mathematical derivations were clear enough.

But I feel like not enough was made of the mixture in the latent space.

When is it expected to be necessary or especially helpful?

Why not use more components in the mixture than 2?

This aspect of the derivation didn't seem fully developed.

What pushed this paper's evaluation down for me were all the tricky details glossed over in the experiments section.

Why not learn the locations of the inducing points?

What control variates were used, and why were they necessary?

In Figure 1 left, why does sparsity level 1.0 for the MoG models have slightly worse performance than the more sparse conditions?

Also, a bit of discussion of deep GPs might have been nice too.

How does this relate to the work by Damianou, Hensman and Laurence?

I really liked the MNIST experiment.

But how were the training hyperparameters tuned?

How does your error rate compare to non-GP methods?

Also, where is the conclusion?

Originality:

This paper is somewhat incremental, in that it combines two existing ideas.

Significance: If the training procedure isn't too fiddly (it looks kind of fiddly), then this method might be a contender for one of the standard things to try on a large dataset, especially for weak likelihoods like in the log-cox model.
Summary: A solid improvement on a nice inference framework.

A bit incremental.

Experiments are relatively well-done but the method looks like it might be hard to use.

Submitted by Assigned_Reviewer_3

This paper takes the pre-existing Gaussian Process inference method of "Automatic Variational Inference" and adapts it for scaling to larger datasets. Specifically, it draws on the method of inducing points to develop a new variational bound that factorises across the data, enabling the use of stochastic optimisation. It uses a mixture of Gaussians to generate approximate posteriors, without being tailored to a specific likelihood model.

I think this article represents a very good contribution to the field: previous work on non-Gaussian likelihoods is generally either an expensive Monte Carlo method, likelihood-specific, or stuck with O(n^3) computational bound. This work represents a method to perform scalable and accurate inference for generic likelihood models: a previously unachieved target that this work makes good progress towards.

The only strong objections I have to the presentation of the work is the representation of experimental results. There is no comparison to competing methods: it would be greatly of interest to compare against "Scalable variational Gaussian process classification." for binary classification, although for other likelihoods there are no clearly competing methods at this scale, so their absence is fair. There is also no timing data: without indication of how quickly the method takes to work, readers will not be able to make a judgement of the accuracy/time tradeoff involved. Ideally, there would be a figure plotting average test error against training time, so the model's learning behaviour can be elucidated. This should be achievable, however, by slight modification to existing code.

Some comments working through the paper:

The introduction is good: the preceding work on Automatic Variational Inference is presented well, as is the method of inducing variables.

Introduction of uncoupled latent Gaussian processes is presented sensibly, along with iid likelihood properties.

The scalability modifications to AVI, drawing on inducing point framework is presented well. Calculation of the of each term is presented well and very thoroughly, to a degree might make as much sense in a supplement.

The introduction of stochastic optimisation is interesting, and could possibly be made more of: unbiased subsampling of data is a hugely useful property for large datasets, and competitors to existing SVI methods would be a good contribution to the literature.

The experimental section is impressive, containing some results that are unprecedented in the literature. As mentioned above, it would greatly benefit from the presentation of timing results, and with comparison against existing methods for scalable Gaussian Process Binary Classification.

A separate conclusion would be valuable to reiterate the main contributions of the work, and the related work section could possibly be moved to earlier in the article.

The language is consistently clear, and the references are well-presented. The supplementary material is accurate and complete.

----- after author rebuttal -----

After the authors' rebuttal, I would be happy to see this paper accepted, if the points raised are fulfilled, i.e. comparison to SVIGPC, timing data and a full conclusion.
Summary: I was quite impressed by this paper: a generic scalable inference technique for Gaussian Process models with non-specific likelihood functions. The results are impressive: my only objection would be to include timing data in the experimental section to demonstrate the method's speed.

Submitted by Assigned_Reviewer_4

Automated variational inference (AVI) is a black-box approach to perform variational inference in Gaussian processes with a variety of likelihoods. One of the drawbacks of the original AVI method was the computational complexity, cubic in the number of samples. In this paper the authors propose a sparse version of AVI that exploits work in the GP community on sparse approximations based on inducing points.

Under the sparse approximation chosen by the authors (and used before in the sparse GP literature), the expected log likelihood can be approximated simply by computing expectations of univariate Gaussian distributions. Importantly, the expected log likelihood term factorizes across samples, making it possible to perform stochastic optimization.

Quality and clarity: This paper is extremely well written and thought out. The experiments are convincing and comprehensive.

Originality: The connection between the original AVI method and the large body of work around sparse variational approximations in GPs is relatively straightforward, but it's good that this paper made it.

Significance: A significant shortcoming of the AVI paper was computational complexity, and this paper convincingly addresses it. I expect this method to be of interest to the NIPS community.

Summary: This is a well written paper that addresses a key problem of automated variational inference. It allows for flexible inference in a large class of models even when dealing with large datasets.

Author Feedback
Author rebuttal: We thank the reviewers for their feedback. Specific comments below.

*R1*

1) Statistical efficiency. This refers to the fact that the expected log likelihood and its gradients can be estimated using expectations over univariate Gaussians, which contrasts with naive estimators that, in principle, would require a larger number of samples from the highly coupled high-dimensional posterior.

2) O(M^3) complexity. The dominant term is the inversion of the covariance on the inducing points. As the objective function and its gradients decompose over the N data-points (see e.g. Eq 17), stochastic optimization can be readily applied, hence the independence on N.

3) Extension to multiple latent functions. This is an important characteristic of our model as our prior considers Q independent GPs that are not necessarily identically distributed, i.e. we allow for different covariance functions, which increases the flexibility of our approach.

4) Mixture approximation. A mixture on the inducing variables yields a mixture on the approximate posterior over the latent functions (Eq 10), which is useful when e.g. dealing with multi-modal posteriors. The experiments only considered up to K=2 following the setup and findings of [6]. Th 1 carries over to Eq 12, hence being fundamental to the mixture approximation as well.

5) Tricky experiments. Most of the experimental set up was defined to allow for a one-to-one comparison with [6] so as to evaluate the main contribution regarding sparsity. The main differences are between the small-scale experiments and large-scale experiments but the details were consistent within each set. Control variates (CV) are necessary to reduce the variance of the gradient estimators and the CV technique was the the same as that in [6]. There are not significant differences across methods on Figure 1 left.

6) Inducing inputs. In order to reduce the overhead of optimizing many more parameters, these have been set in our experiments using clustering (as described in the supplementary material). However, preliminary experiments indicate that similar performance can be achieved with a much smaller number of inducing inputs when these are learned along with all other parameters.

7) Deep GPs. Deep GPs are mainly focused on hierarchical GPLVM models and do not deal with general likelihoods (as presented at AISTATS 2013). We will discuss these models in the final version of the paper.

8) MNIST experiment. Hyperparameters are learned by optimization of the variational objective. Previous literature reports that linear classifiers achieve 12% error while state-of-the-art methods based on large/deep conv. nets obtain less than 1%. Our results show that our method, while solving the harder problem of full posterior estimation, can reduce the gap between GPs and deep nets.

9) Conclusion. We agree and will make space for a separate conclusion reiterating main contributions.

10) Fiddly training. Our framework is easy to use and can be applied successfully with default settings as most details have been taken care of in our implementation, see 5) above.

*R2*

1) Comparison to [22]: After running our framework on the same MNIST binary classification experiment as in [22], we obtain an accuracy of 97.8% and an NLP of 0.068, which are very similar to those those reported by [22], 97.8% and 0.069, respectively.

2) Timing data. We thank the reviewer for this suggestion. In the MNIST experiment above, we observe a rapid decrease of the error rate from around 50% to 10% in the first 1000 secs and then converging to near best predictions after 2000 secs.

3) Conclusion. See R1, 9 above.

*R7*

1) Inducing points. See R1, 6 above.